# Impact of Coronavirus Disease COVID-19 on the Relationship between Healthcare Expenditures and Sustainable Economic Growth

**DOI:** 10.3390/ijerph20043049

**Published:** 2023-02-09

**Authors:** Alina Vysochyna, Tetiana Vasylieva, Oleksandr Dluhopolskyi, Marcin Marczuk, Dymytrii Grytsyshen, Vitaliy Yunger, Agnieszka Sulimierska

**Affiliations:** 1Academic and Research Institute of Business, Economics and Management, Sumy State University, 40007 Sumy, Ukraine; 2Department of Applied Social Sciences, Silesian University of Technology, 44-100 Gliwice, Poland; 3Faculty of Economics and Management, West Ukrainian National University, 46020 Ternopil, Ukraine; 4Institute of Public Administration and Business, WSEI University, 20-209 Lublin, Poland; 5Faculty of National Security, Law and International Relations, Zhytomyr Polytechnic State University, 10005 Zhytomyr, Ukraine; 6Faculty of Management, Lublin University of Technology, 20-618 Lublin, Poland

**Keywords:** coronavirus disease, COVID-19, healthcare expenditure, pandemic, sustainable economic growth, sustainable development

## Abstract

The coronavirus disease (COVID-19) pandemic led to a catastrophic burden on the healthcare system and increased expenditures for the supporting medical infrastructure. It also had dramatic socioeconomic consequences. The purpose of this study is to identify the empirical patterns of healthcare expenditures’ influence on sustainable economic growth in the pandemic and pre-pandemic periods. Fulfilment of the research task involves the implementation of two empirical blocks: (1) development of a Sustainable Economic Growth Index based on public health, environmental, social, and economic indicators using principal component analysis, ranking, Fishburne approach, and additive convolution; (2) modelling the impact of different kinds of healthcare expenditures (current, capital, general government, private, out-of-pocket) on the index using panel data regression modelling (random-effects GLS regression). Regression results in the pre-pandemic period show that the growth of capital, government, and private healthcare expenditures positively influence sustainable economic growth. In 2020–2021, healthcare expenditures did not statistically significantly influence sustainable economic growth. Consequently, more stable conditions allowed capital healthcare expenditures to boost economic growth, while an excessive healthcare expenditure burden damaged economic stability during the COVID-19 pandemic. In the pre-pandemic period, public and private healthcare expenditures ensured sustainable economic growth; out-of-pocket healthcare expenditures dominantly contributed to the pandemic period.

## 1. Introduction

In 2022, a violation of national security, macroeconomic stability, and sustainable economic growth resulted from the spread of the COVID-19 pandemic and the emergence of new threats (monkeypox, escalation of military conflicts, etc.). In particular, according to the International Monetary Fund forecasts [1], the global economic growth rate in 2022 will slow down from 6.1% to 3.2%. It is even lower than a similar indicator from April 2021. Experts also note that there remains a high probability that most economic, environmental, and food risks will be realised. Therefore, an increase in inflation is expected. As a result, the global economic growth will probably decrease to approximately 2.6% at the end of 2022, and to 2.0% in 2023, which is the lowest growth rate since 1970. In these conditions, government officials worldwide are involved in developing a new paradigm of socioeconomic policy to maintain the state apparatus’ relatively stable functioning and prevent a further deterioration of households’ well-being. It underlines the importance of reforming fiscal and budgetary policy (in particular, the restructuring and optimisation of public expenditures) to ensure the recovery of sustainable economic growth dynamics. In this context, determining the empirical patterns of the healthcare expenditures’ impact on sustainable economic growth, as well as identifying the transformational patterns of this causality in pandemic and pre-pandemic periods becomes particularly relevant. It forms the main objective of the research. The main objective of the research considers the fulfilment of several tasks: (1) conducting trend and bibliometric analyses of Scopus publications on healthcare expenditure and sustainable growth to underline key contextual vectors of scientific research in this subject area and their evolution; it also might become a theoretical background for measurement indicator selection; (2) formation of the Sustainable Economic Growth Index (SEGI) as a complex dependent variable in further empirical research; (3) modelling the impact of healthcare expenditures on the Sustainable Economic Growth Index in the pre-pandemic (2000–2019) and pandemic (2020–2021) periods. The core assumption of the study is that COVID-19 triggered a transformation of healthcare expenditure relevance in boosting sustainable economic growth. Namely, in the pre-pandemic period, there was no significant healthcare expenditure burden on the country’s fiscal sustainability. Otherwise, in times of coronavirus disease, healthcare systems are considerably depressed by the number of ill patients and the scale of medical invasions required to save patients’ lives. During a pandemic period, the efficiency of a healthcare system influences labour force quality and sufficiency as well as household well-being. Considering all those mentioned above, an assumption has developed that healthcare expenditure might play a more significant role in promoting sustainable economic growth in more turbulent periods (pandemic) than in more stable periods (pre-pandemic).

It should be noted that this statement is quite fair only for high- and middle-income countries but not for low-income countries. In [2], scientists analyse how the public health system influences environment, economic development, and social inequality in low-income countries. The authors note that environmental threats in developed and emerging countries differ significantly. First, in emerging countries, environmental threats are determined by food insecurity, climate risks, availability of water resources, sanitary conditions, etc. In turn, all these factors damage public health. Consequently, solving environmental problems for emerging countries has a high contribution to public health and sustainable development. Thus, it is better to run the empirical research using data from a country sample with similar economic development trends. Otherwise, considering both developed and emerging countries within the integral empirical research might lead to unreliable results.

Considering this, the country sample of the research consists of 15 European countries, including Albania, Bulgaria, Croatia, Czech Republic, Estonia, Hungary, Latvia, Lithuania, Moldova, Poland, Romania, Serbia, Slovakia, Slovenia, and Ukraine. The country sample was created considering both geographical proximity to Ukraine and Poland, as well as the similarity of economic development trends (in particular, purchasing power standards and income). The period of observations is 2000–2021.

## 2. Theoretical Background

Bibliometric analysis was carried out with VOSviewer_1.6.17 toolkit [3] to clarify the general contextual directions of scientific research aimed at identifying the relationship between healthcare expenditure and sustainable economic growth. This theoretical block of research was realised based on 1294 Scopus [4] papers, the title, abstract, or keywords of which contain such words as “health”, “expenditure”, and “sustainable”.

It is worth noting that the first publication [1] that meets the search request was published in 1990. Thus, the research covers a period of 1990–2022. The dynamics of publication activity on healthcare expenditure and sustainable growth in 199–2022 are presented in Figure 1.

According to Figure 1, several stages of publication activity concerning healthcare expenditure and sustainable growth can be distinguished, namely:1990–2005—the germinal stage that is characterised by a low intensity of publication activity (on average six articles on the topic were published annually);2006–2015—the stage is characterised by a moderate intensity of publication activity (on average 35 articles on the topic were published annually);2016–2021—the stage of high intensive publishing activity (on average 120 articles on the topic were published annually).

It is also worth noting that as of 15 September 2022, 140 scientific papers on healthcare expenditure and sustainable growth have already been published. However, based on the forecasting results using the exponential trend line (it was chosen among other alternative options taking into account the value of the coefficient of determination), the number of publications on healthcare expenditure and sustainable growth by the end of the year should exceed 183 articles. Thus, the presented analytical information convincingly testifies to strengthening the scientific community’s attention to the specified problem. This trend significantly intensified after 2016, which may also be due to the approval of the Sustainable Development Goals. The COVID-19 pandemic also triggered an almost 30% growth in the relevant Scopus publications (in 2020 compared to 2019).

At the same time, Figure 2 illustrates dissemination by the residential status of academicians who have been researching the relationship between healthcare expenditure and sustainable growth.

According to Figure 2, it can be noted that leadership in the study of the relationship between the financing of the healthcare system and sustainable development is held by the USA, whose researchers published 335 publications on the topic during 1990–2022. Scientists from the United Kingdom, China, and India also have a high level of publication activity. At the same time, researchers from Australia, Canada, the Netherlands, South Africa, Italy, and Germany have approximately the same number of publications.

Figure 3 reflects the co-occurrence between the key concepts that scientists highlight in their research to study the relationship between healthcare expenditure and sustainable growth.

Thus, according to Figure 3, seven contextual clusters of scientific research, which cover 933 keywords, can be identified, namely:red cluster (268 keywords)—covers scientific research focused on studying the impact of environmental, energy, food, and urban factors on sustainable economic development, population health, and healthcare expenditures, and the impact of coronavirus disease (COVID-19) on the dynamics of sustainable growth and scale of financing the healthcare system;green cluster (226 keywords)—covers scientific research focused on evaluating the effectiveness of healthcare institutions, the mechanism of medical services’ cost formation, and optimisation of healthcare expenditures;blue cluster (136 keywords)—covers scientific research on the impact of age, gender, and anthropogenic factors (harmful habits, unhealthy lifestyle, etc.) on population health;yellow cluster (103 keywords)—covers scientific research on finding mechanisms for reducing child mortality and improving maternal care;purple cluster (91 keywords)—covers scientific research on the influence of the medical services’ availability and the cost of drugs on the formation of the cost of medical services;turquoise cluster (65 keywords)—covers scientific studies focused on assessing the impact of chronic diseases, household income, social status, and family size on household well-being and health;orange cluster (43 keywords)—covers scientific studies focused on determining the impact of the spread of HIV, malaria, and tuberculosis on the volume of healthcare expenditures.

Thus, it can be noted that most clusters are focused on the study of sustainable development issues or healthcare financing issues. At the same time, there is a lack of interdisciplinary research. The only interdisciplinary cluster is red.

Evolutionary patterns of changes in the research on identifying the relationship between healthcare expenditures and sustainable growth can be carried out considering data from Figure 4.

Most of the current scientific research on the topic is related to the coronavirus pandemic, progress in fulfilling the Sustainable Development Goals, reducing greenhouse gas emissions and improving the quality of the environment (especially the quality of air), the effectiveness of various models of financing healthcare (in particular, universal life insurance), improving the education of the population in the field of public health. At the same time, earlier publications are related to the impact of the availability of medical services and the quality of administration of the medical institutions in terms of ensuring the effectiveness of the public health system and the impact of environmental and resource factors on sustainable development.

Based on the generalisation of the results of the bibliometric analysis, it can be noted that modern research on the relationship between healthcare expenditures and sustainable development is determined mainly by the coronavirus disease COVID-19 pandemic and the need to achieve targets within the framework of the Sustainable Development Goals. Nevertheless, there is a need for deepening interdisciplinary research that combines healthcare and sustainable development issues. This confirms the relevance of this study aimed at determining the empirical patterns of the impact of healthcare expenditures on sustainable development, as well as identifying the transformation of these patterns during the pandemic period.

### 2.1. Sustainable Economic Growth Measurement: Literature Review

A more detailed literature review is required to formalise the indicators of a quantitative assessment of sustainable economic growth, as well as to identify the tools for determining the impact of healthcare expenditures on sustainable economic growth.

It can be noted that approaches to determining the parameters of a quantitative assessment of sustainable economic growth are significantly differentiated. In particular, in the earliest publication on this topic [5], published in Scopus [4], researchers identified gross national product as a key indicator of economic development measurement, and also analysed the change of this indicator under the influence of environmental determinants. The scientists concluded that the traditional view of the gross national product’s dependence on the dynamics of economic growth is not entirely fair, because environmental risks and losses remain neglected. Finally, the researchers emphasised that sustainable economic growth should be based on the need to implement environmental expenditures on preventive, restorative, and compensatory measures (“defensive expenditures”).

In one paper [5], the relationship between healthcare expenditure and economic growth in developing countries (Czech Republic, Egypt, Greece, Hungary, Poland, Russian Federation, South Africa, UAE, China, India, Indonesia, Republic of Korea, Malaysia, Philippines, and Thailand) in 1995–2013 is investigated. GDP per capita is chosen in the research as a proxy of economic growth.

Authors of [6] realised a study on revealing both linear and nonlinear causality between healthcare expenditure and economic growth for fifteen Organisation for Economic Co-operation and Development member states and five major developing countries in 1995–2015. They also chose GDP as a proxy of a country’s economic development.

The authors of [7] investigate the short-run and long-run relationship between healthcare expenditure and economic growth considering only the Turkish case for 1980–2015. It identified GDP and investment as measurement indicators of economic growth.

In [8], pairwise causality is explored within the triangle “healthcare expenditure–economic growth–environment” for Taiwan over the period 1995 Q1–2016 Q4. It used real GDP per capita as a proxy of economic growth and CO_2_ emission as a proxy of the environmental efficiency. Authors argued that there are significant lead–lag relationships between indicators in the triangle. It proves the necessity of a consideration of environmental determinants when identifying the causality between economic growth and healthcare expenditure.

While most of the research mentioned above considered economic growth and sustainability within the traditional growth model approach, in the papers [9,10,11,12,13,14,15,16], a country’s economic development is characterised through a complex and systemic approach using the following indicators: central government debt, total % of GDP; GDP growth, annual %; GDP per capita, PPP constant 2011 international USD; gross fixed capital formation, % of GDP; income share held by the lowest 20%; industry value added, annual % growth; research and development expenditure, % of GDP; unemployment, % of the total labour force; current account balance, % of GDP [17,18].

Authors of [19], while researching healthcare expenditure and economic growth causality for South Asian Association for Regional Cooperation countries by employing the Panel cointegration and panel causality analysis over the period 1995–2012, used GDP per capita, labour force, literacy rate, and elderly population of age 65 as measurement indicators of economic development.

### 2.2. Healthcare Expenditure Optimisation: Literature Review

It should be noted that most of the modern research on healthcare expenditures is aimed at searching for ways of optimisation and finding mechanisms for obtaining greater outcomes for fewer inputs. It should also be noted that there are no standard solutions for low-income and high-income countries because there are significant differences in the cost of medical services and their quality and urgency [5,20,21].

In [22], the patterns of changes in the basic principles of the organisation of healthcare systems over the past 25 years in low-income and middle-income countries are analysed. The researchers emphasise that in this group of countries, the priority by 2030 should be ensuring the availability of surgical services, the timeliness and quality of their provision. Scientists note that developing this particular segment of medical services will significantly reduce healthcare expenditures. Specifically, they pointed out that “provision of safe and affordable surgical care when needed not only reduces premature death and disability, but also boosts welfare, economic productivity, capacity, and freedoms, contributing to long-term development”.

Authors of [23,24] pointed out the importance of sufficient food security in promoting sustainable development and ensuring public health. The researchers in [25] also analysed the relationship between food factors, sustainable development, and public health. Scientists analysed the prospects of changing the diet of women aged 19–50 in favour of products that require less greenhouse gas emissions. According to the analysis results, it is confirmed that it is pretty realistic to change the approach to nutrition, which will simultaneously reduce the negative impact on the environment and at the same time not harm health.

In [26], the sustainable development perspective of the healthcare system in the USA is examined from the standpoint of balancing both public and private expenditures in this sphere. In particular, the researchers determined that more rational use of medical consumables and medical waste reduction will significantly reduce unproductive expenses. According to preliminary estimates by researchers, savings from such innovations can amount to 20% of total healthcare expenditures. 

Optimising healthcare expenditures (Medicare) has also been the focus of scientific research [27]. By analysing healthcare expenditures for patients with chronic diseases within 15 healthcare programs during 2002–2006, researchers concluded that improvements in care, patient adherence, and communication could help reduce the cost burden on hospitals and Medicare.

Researchers [28] also analysed the optimisation of healthcare expenditures from the perspective of timely prevention and reducing the risks of Chagas disease spreading.

In [29], healthcare expenditures have also been researched. The authors highlighted that making essential medicines affordable might help to ensure health security. Scientists’ recommendations highlight the necessity of governments or healthcare systems covering essential medicines packages and reducing out-of-pocket expenditures.

Scientists in [30], based on data analysis for 25 EU countries in 2005, determined the factors of the life expectancy increase in these countries and healthy life years among the population over 50 years old. The researchers found that economic growth, lifelong education, and financing of elderly care are the drivers of healthy life years growth in these countries. At the same time, long-term unemployment inhibited it.

Thus, in [31], the impact of healthcare expenditures on mortality rates and life expectancy is analysed, based on data for 25 selected sub-Saharan African countries from 2000 to 2020. Using a panel spatial correlation consistent approach, authors revealed that increased healthcare expenditures lead to reduced infant and maternal mortality and positively affect life expectancy. Moreover, the authors of [32] analysed the relationship between health insurance and maternal mortality in 10 West African countries. Based on empirical research results, authors concluded that participation in health insurance schemes helps people to gain sufficient access to primary medical services during pregnancy, which reduces maternal mortality. These researchers concluded that an expansion of both government and private healthcare expenditures leads to an improvement of life expectancy and the elimination of mortality risks in low- and middle-income African countries.

An interesting study [33] based on data from Tanzania revealed patterns and consequences of expanding the network of mandatory health insurance coverage for low-income countries. According to the results of the analysis, it was established that expanding the coverage network could negatively affect the financial stability of institutions that serve such applicants. At the same time, scientists have proven that the structure of medical services and their cost varies significantly in low- and high-income countries, which determines the feasibility of conducting similar studies with wider geographical coverage. At the same time, the interim conclusions indicate the need for a balanced approach to the rapid growth of mandatory health insurance because healthcare institutions may not be ready for such a load.

Under fiscal decentralization, it is important to also identify the public health impact on sub-central governments’ healthcare expenditures. Thus, scientists [34] studying the Chinese experience found out that the growth of sub-central governments’ financial autonomy has a positive effect on the efficiency of healthcare expenditure allocation.

Thus, in order to realise reliable empirical research on the identification of the impact of healthcare expenditures on sustainable economic growth, several essential preconditions might be considered: (1) a geographic sample might consist of countries with familiar economic development trends; (2) it is better to test the impact of different types of healthcare expenditures (public, private, out-of-pocket, etc.); (3) it is better to follow a multidimensional approach for sustainable growth measurement, which cover economic, environmental, and social perspectives despite using only the GDP per capita.

The next block of theoretical research Is focused on the formalisation of the channels of the influence of coronavirus disease COVID-19 on various components of sustainable socioeconomic development. Thus, scientific studies [35,36] are focused on assessing the impact of COVID-19 on sustainable economic growth through the channel of population employment and labour migration. A significant array of scientific publications [37,38] are also devoted to analysing the changes in the organisational and functional parameters of business models. Scientists note that government support for businesses most damaged by the pandemic (tourism, hotel, and restaurant business, etc.) is an essential precondition for post-pandemic economic recovery. The authors of [38] also found that the most widely used instruments of state business support during the post-pandemic recovery are “debt financing, employment support, tax benefits, and reduction of administrative influence on business”. A specific block of research [39,40] focused on assessing healthcare sector efficiency in counteracting COVID-19 negative consequence.

Summarising the results of the theoretical analyses, it is worth noting that the evolution of scientific work on determining the parameters of a quantitative assessment of sustainable economic growth has a more than 30-year history. Researchers identify as proxies of economic growth quantification such indicators as GDP growth, unemployment/employment rate, inflation, investment activity, current account balance, trade turnover, etc. At the same time, scientists note that achieving sustainable economic growth is impossible without considering social and environmental determinants.

In turn, the COVID-19 pandemic significantly adjusted algorithms and models for ensuring sustainable economic growth. This was related to fiscal and budgetary policy because the need for additional financing of healthcare institutions became acute. In this regard, the determination of the empirical patterns of the impact of healthcare expenditures on sustainable economic development, as well as the identification transformation of these patterns in pandemic period, is relevant.

## 3. Materials and Methods

Fulfilment of the research task involved the implementation of two empirical blocks, namely: (1) the formation of a sustainable economic growth integral indicator; (2) modelling the impact of healthcare expenditures on the Sustainable Economic Growth Index (SEGI) in the pre-pandemic period and in times of the coronavirus disease spread.

### 3.1. Sustainable Economic Growth Index

An important stage in the formation of the Sustainable Economic Growth Index is identification of those indicators that most accurately and comprehensively allow us to assess the progress in sustainable economic growth. In particular, it is worth noting that within the framework of this study, considering sustainable economic growth through the Sustainable Development Goal No. 8 “Decent Work and Economic Growth” is proposed. Thus, sustainable economic growth depends on sufficient levels of employment, GDP growth, and sufficient volume of investments and their redistribution in promising areas, ensuring an acceptable level of personal income, a levelling of social inequality, a transition to a low-carbon economy, and efficient trade. Considering all above-mentioned and literature review results, 13 indicators of sustainable economic growth are chosen, including the following:SEG_1—Life expectancy at birth, total (years);SEG_2—Unemployment, total (% of total labour force) (modelled ILO estimate);SEG_3—Gini index;SEG_4—CO_2_ emissions (metric tons per capita);SEG_5—Electric power consumption (kWh per capita);SEG_6—GDP growth (annual %);SEG_7—Foreign direct investment, net inflows (% of GDP);SEG_8—Gross capital formation (% of GDP);SEG_9—High-technology exports (% of manufactured exports);SEG_10—Medium- and high-tech manufacturing value added (% manufacturing value added);SEG_11—Inflation, consumer prices (annual %);SEG_12—New business density (new registrations per 1000 people ages 15–64);SEG_13—Trade (% of GDP).

All these indicators are accumulated for 15 European countries for 2000–2021 from the World Bank’s World Development Indicators collection [41]. It is worth noting that unemployment, total (% of total labour force) is a measure of labour market development. Economic growth is traditionally characterised by GDP growth (annual %), as well as inflation, consumer prices (annual %) and new business density. The sufficiency of investment provision is proposed to be measured by such indicators as foreign direct investment, net inflows (% of GDP), gross capital formation (% of GDP), medium- and high-tech manufacturing value added (% manufacturing value added). Progress in levelling social inequality might be monitored with the Gini index. Progress in the transition to a low-carbon economy should be monitored based on CO_2_ emissions (metric tons per capita) and electric power consumption (kWh per capita). The trade sufficiency can be evaluated through such indicators as high-technology exports (% of manufactured exports) and trade (% of GDP). Life expectancy at birth is an additional integrating element. Sustainable economic development determines citizens’ well-being and quality of life, which inevitably affects life expectancy.

Descriptive statistics for selected indicators of sustainable economic growth are presented in Table 1.

As can be seen from the descriptive statistics, the dataset contains omitted observations, and therefore these observations were forecasted with an extrapolation technique. In addition, it can be seen from Table 1 that the selected indicators are not commensurate, which determines the need to bring them to a comparable form by normalisation. We propose normalising indicators using the method of relative normalisation. In the process of normalisation, the variables SDG_2–SDG_5 and SDG_11 are considered inhibitors of sustainable economic growth, while the remaining indicators are drivers of sustainable economic growth. The application of the relative normalisation method will allow us to concentrate the normalised values of indicators in the range [0; 1], where a larger value of the indicator clarifies its higher positive role in ensuring sustainable economic growth.

The next stage of the research considers using the principal component analysis in Stata 12/SE software, ranking with the built-in function in MS Excel software and Fishburne scheme in order to clarify weightings for the coefficients.

An integral indicator using additive convolution based on Formula (1) is proposed:(1)$$SEGI=\overset{n}{\underset{i=1}{\sum }}w_{i}q_{i}$$
where SEGI—Sustainable Economic Growth Index; $w_{i}$—weighting coefficient of the individual measurement indicator of sustainable economic growth; $q_{i}$—the actual value of the individual measurement indicator of sustainable economic growth.

### 3.2. Identification of Healthcare Expenditure Impact on Sustainable Economic Growth

The next stage of this study involves modelling the impact of healthcare expenditures on the Sustainable Economic Growth Index in the pre-pandemic and pandemic periods to determine the specific patterns of this relationship and their transformation in the times of the coronavirus.

In order to investigate causality between economic development and healthcare expenditure, scientists used numerical econometric instruments (ARDL, GMM, OLS regression, Granger causality test, panel-type VECM model, etc.), but in the paper, a unified econometric instrument is proposed—panel data GLS regression and Hausman test as is also proposed in [17]. So, the panel data regression modelling toolkit will be used to fulfil this task. The functional form of the relationship (fixed or random effects model) that most accurately fits the given data sample will be determined using the Hausman test.

The dependent variable is the Sustainable Economic Growth Index. In most of the analysed papers, general healthcare expenditure per capita is used as a single and unified measurement indicator. Nonetheless, in this study, the use of different kinds of healthcare expenditure is proposed in order to obtain more specific modelling results:Capital healthcare expenditure (% of GDP) (CapHE);Current healthcare expenditure (% of GDP) (CurHE);Domestic general government healthcare expenditure (% of general government expenditure) (DGGHE);Domestic private healthcare expenditure (% of current healthcare expenditure) (DPHE);Out-of-pocket expenditure (% of current healthcare expenditure) (OoPE).

All statistical data were generated from the collection “Health Nutrition and Population Statistics” of the World Bank [41].

It should be noted that two regression models will be built. Model 1 covers the pre-pandemic period—2000–2019, while model 2 covers the period of the pandemic—2020–2021. Comparing the obtained modelling results will allow specific patterns of influence of different healthcare expenditures on sustainable economic growth to be determined.

## 4. Results

The main task of this study was to determine the empirical patterns of the impact of healthcare expenditures on sustainable economic development, as well as to identify transformations of these patterns during the pandemic period. The practical implementation of the task involves two stages, the results of which are presented below.

Based on the results of the principal component analysis, it is determined that five principal components explain 70.32 of the total sustainable economic growth indicators’ variation. Thus, determining the importance of individual indicators on the formation of the Sustainable Economic Growth Index is achieved using the average eigenvalues within these five principal component eigenvectors (columns PCE_1–PCE_5, Table 2). This indicator (column Average, Table 2) is the basis for further ranking of indicators using the corresponding built-in function of MS Excel (column Rank, Table 3). It is established that two indicators—SEG_4 (CO_2_ emissions (metric tons per capita)) and SEG_5 (Electric power consumption (kWh per capita))—have the same average eigenvalue and therefore the relevance of these individual indicators was assessed with the same rank—9. The sum of all ranks for 13 individual indicators of sustainable economic growth is 90. Thus, the weighting coefficients of the Sustainable Economic Growth Index (column WtC, Table 2) are calculated by the ratio of the rank assigned to a specific individual indicator to the sum of all ranks.

Table 3 shows that such indicators as SEG_6—GDP growth, SEG_7—foreign direct investment, net inflows and SEG_2—unemployment have the most significant importance in ensuring sustainable economic growth in the 15 European countries. In addition, these individual indicators are closely connected to the targets of the Sustainable Development Goal No. 8 “Decent Work and Economic Growth”, which additionally confirms the validity of the author’s approach. In turn, the least relevant among the selected indicators in the context of ensuring sustainable economic growth are trade (SEG_13), medium- and high-tech manufacturing value added (SEG_10), and life expectancy at birth (SEG_1).

Considering the weighting coefficients presented in Table 2 and the normalised values of individual indicators, the Sustainable Economic Growth Index was calculated using Formula 1. SEGI calculation results for 15 European countries are presented in Table 3.

According to Table 3, it can be noted that in the 15 selected European countries, the potential for sustainable economic growth is only half used because the level of the Sustainable Economic Growth Index during 2000–2021 fluctuates in the range [0.221; 0.511] with the maximum value of the indicator 1. It is worth noting that the minimum level of the SEGI was reached in 2009 in Lithuania, while the maximum was reached in 2020 in Hungary. Hungary, Moldova, and Estonia have the highest level of Sustainable Economic Growth Index. Ukraine, Bulgaria, Croatia, and Serbia are outsiders according to this indicator.

The functional form of dependence between these variables is determined using Hausman test in the Stata 12/SE software product. According to the test results, it was determined that “Prob>chi2 = 0.3390”. The results indicate the need to reject the null hypothesis and accept the alternative one. Thus, the form of the panel data regression model with random effects is more acceptable.

Two models were built within this block of the research: model 1 (Table 4) characterises the impact of healthcare expenditures on sustainable economic growth in 15 European countries in the pre-pandemic period, and model 2 (Table 5) reflects these relationships during the pandemic period.

Thus, the impact of healthcare expenditures on sustainable economic growth in 15 European countries in the pre-pandemic period (2000–2019) can be characterised as follows:all independent variables have a statistically significant effect on the Sustainable Economic Growth Index at one of the acceptable confidences intervals;an increase of 1% in the capital healthcare expenditure to GDP ratio will lead to the growth of the Sustainable Economic Growth Index at 0.05025 units with a confidence probability of 99%;the Sustainable Economic Growth Index is also positively affected by the increase in the domestic general government healthcare expenditure to general government expenditure ratio and the domestic private healthcare expenditure to current healthcare expenditure ratio, namely, a 1% increase in the independent variable leads to an increase in the dependent variable by 0.01543 and 0.00509 units, respectively, with a 99% confidence probability;on the contrary, 1% increase in the current healthcare expenditure to GDP ratio will lead to a decrease in the Sustainable Economic Growth Index at 0.01303 units with a 99% confidence probability;a 1% increase in the out-of-pocket expenditure to current healthcare expenditure ratio with a probability of 90% will lead to a decrease in the Sustainable Economic Growth Index by 0.00258 units.

Thus, sustainable economic growth in 15 selected European countries in the pre-pandemic period is ensured by capital healthcare expenditure growth. The scale of this indicator’s impact on the dependent variable is the largest in the model. In turn, current healthcare expenditures have the most significant negative impact on sustainable economic growth in the pre-pandemic period. Thus, it is quite natural to conclude that the contribution to sustainable economic growth becomes noticeable only under the development of the healthcare infrastructure and capital repair, modernisation, and re-equipment of existing facilities. In contrast, the growth of current healthcare expenditure allows only for current needs to be covered without investing in future growth.

The analysis of the regression modelling results to determine the impact of healthcare expenditures on the Sustainable Economic Growth Index for 15 European countries in the pandemic period (2020–2021), presented in Table 5, allows us to draw the following conclusions:all independent variables are characterised by an absence of statistical significance of the impact on the Sustainable Economic Growth Index at any of the acceptable confidence intervals, which can be explained by the small number of years of observation;the model shows that during the pandemic period, growth in capital healthcare expenditure and current healthcare expenditure, as well as domestic private healthcare expenditure has a negative impact on sustainable economic growth;on the contrary, a 1% increase in the domestic private healthcare expenditure to current healthcare expenditure ratio and the out-of-pocket expenditure to current healthcare expenditure ratio can lead to an increase in the Sustainable Economic Growth Index by 0.00222 and 0.00172 units, respectively.

Thus, the modelling results proved that in the time of coronavirus disease COVID-19, the patterns of healthcare expenditure impact on the Sustainable Economic Growth Index for 15 European countries are changed in comparison with the pre-pandemic period. In particular, in more stable socioeconomic conditions, a promising vector for ensuring stable economic growth is an increase in healthcare capital expenditures. At the same time, in the time of coronavirus disease, which is characterised by an excessive burden on the healthcare system, the increase in both current and capital expenditures create serious risks, not only of a violation of macroeconomic stability and a threat to national security but also a threat to sustainable economic growth.

## 5. Discussion

Realisation of the research objectives has resulted in two blocks of findings.

In particular, the results of the bibliometric analysis, obtained based on the analysis of 1294 Scopus [3] papers, the title, abstract, or keywords of which contain such words as “health”, “expenditure”, and “sustainable”, testify that the most dynamic period of publishing activity on this topic is 2016–2021. During this period, on average about 120 articles on the topic were published annually. In this period, the focus of scientific research is directed to the study of the coronavirus pandemic, progress in the fulfilment of the Sustainable Development Goals, reducing greenhouse gas emissions and improving the quality of the environment (especially air quality), the effectiveness of various models of healthcare financing (in particular, universal life insurance), and improving the education in the field of public health. In general, all publications can be divided into seven contextual clusters. The largest clusters are focused on the analysis of the role of environmental, energy, food, and urban factors in ensuring sustainable growth (including the impact of the pandemic), evaluating the effectiveness of the healthcare institutions and healthcare expenditures, and identifying the influence of age, gender, anthropogenic factors (harmful habits, unhealthy lifestyle, etc.) on the population health.

A more detailed study of the existing scientific papers in this direction revealed that the leading indicators of sustainable economic growth are GDP growth, unemployment/employment ratio, inflation, investment activity, current account balance, trade turnover, etc. At the same time, achieving sustainable economic growth is impossible without considering social and environmental determinants.

The obtained theoretical results served as the basis for the implementation of this study’s second empirical block. Based on a combination of the relative normalisation method, principal component analysis in Stata 12/SE software, ranking using a built-in function in MS Excel, Fishburne scheme, and additive convolution, the Sustainable Economic Growth Index has been developed. Among the 13 individual indicators of the Sustainable Economic Growth Index, GDP growth, foreign direct investment, net inflows, and unemployment have the most significant importance within this geographic sample (15 European countries). The researched papers also mention these indicators as core proxies of economic development. Based on the results of the calculations, it was determined that during 2000–2021 the Sustainable Economic Growth Index varied in the range [0.221; 0.511]. The countries with a high level of Sustainable Economic Growth Index include Hungary, Moldova, and Estonia, while Ukraine, Bulgaria, Croatia, and Serbia are outsiders according to this indicator.

According to the panel data regression analysis results (random-effects GLS regression) in 2000–2019, it is found that all types of healthcare expenditures have a statistically significant impact on the Sustainable Economic Growth Index. However, the growth of capital expenditures, domestic general government healthcare expenditure, and domestic private healthcare expenditure led to the growth of the Sustainable Economic Growth Index in the pre-pandemic period. These conclusions also correlate with the research results [5,8,11], in which authors show evidence that healthcare expenditure (per capita) helps to maintain an economic growth dynamic (measured by GDP per capita) in developing European, Middle East African, and Asian countries. In [7], it is also empirically proved that there is a positive long-run relationship between economic growth and public and private healthcare expenditure (case of Turkey, 1980–2015). While in [12], it is pointed out that there is positive relationship between personal healthcare expenditure and gross national product in the USA. In the paper [18], the positive influence of healthcare expenditure on economic growth is also confirmed, based on data for 31 countries from 1986 to 2007. In turn, the authors of [21,42] revealed that public and private health expenditure negatively impact economic growth.

On the contrary, current healthcare expenditures and out-of-pocket expenditures negatively affect sustainable economic growth dynamics. It is interesting that in [20], while exploring the linkage between public healthcare expenditure and economic growth for 43 developing countries over 20 years, opposite results were obtained. Basically, authors pointed out that current expenditures positively impact the growth rate dynamic, while an increase in capital expenditure is negative.

In turn, the panel data regression analysis results in 2020–2021 did not reveal a statistically significant impact of healthcare expenditures on sustainable economic growth during the COVID-19 pandemic. In turn, the modelling results showed a negative impact in the COVID-19 period of capital and current healthcare expenditures on the Sustainable Economic Growth Index. However, the impact of domestic general government healthcare expenditure and out-of-pocket expenditure on the sustainability of economic development is positive. Nonetheless, these results can be considered as reliable because the low quality of the model resulted from a lack of observations.

It should be noted that all the above-described modelling results are fair only for the selected 15 European countries, but the methodology of the Sustainable Economic Growth Index’s construction and identification of the impact of healthcare expenditure on sustainable economic growth in pre-pandemic and pandemic periods might be applied to other geographic sample to obtain more complete results. The approach to development of the Sustainable Economic Growth Index might be applied to any country sample because it relies on public data (the World Bank Collections) and its construction is based on a step-by-step approach, which might be adapted to any country sample. Moreover, measurement indicators of sustainable economic growth were selected based on the literature review of differentiated publications, which makes its country-specific dependence relatively low.

## 6. Conclusions

The generalisation of the theoretical block of this study regarding the identification of the relationship between healthcare expenditures and sustainable economic growth proves that scientific research in this direction is closely related to challenges and risks arising in the socioeconomic environment. This fact is confirmed by the results of the overlay visualisation built using the VOSviewer_1.6.17 toolkit [3]. In particular, after the approval of the Sustainable Development Goals, the attention of scientists was focused on the study of the prerequisites and prospects for their implementation and impact on economic growth. In turn, during the spread of the coronavirus disease, researchers began to analyse more intensively the impact of the pandemic on various spheres of socioeconomic life. It is also worth noting that the central part of scientific research is focused on the impact of environmental, energy, food, and urban factors on sustainable economic development, population health, and healthcare expenditures, as well as the impact of coronavirus disease COVID-19 on the dynamics of sustainable growth and the scale of financing the healthcare systems.

The generalisation of the empirical block of this study allows us to conclude that in more stable socioeconomic conditions, ensuring the stability of economic growth implies an increase in capital healthcare expenditures. Capital health expenditures according to the World Bank methodology [41] “include health infrastructure (buildings, machinery, IT) and stocks of vaccines for emergency or outbreaks”. Positive impact of capital healthcare expenditures on sustainable growth might be explained by following a logical chain: the development of medical infrastructure and creating stocks of vaccines for emergencies or outbreaks allows the provision of more efficient medical services; more efficient medical services contribute to the improvement of public health, well-being, and quality of workforce; a more productive workforce contributes to GDP growth. In turn, current health expenditures according to the World Bank methodology [41] “include healthcare goods and services consumed during each year”. Therefore, this type of expenditure allows the operational activity of healthcare institutions to be supported. Its negative impact on sustainable economic growth might illustrate an insufficient expenditure structure (excessive current expenditure and ineffective allocation of budget resources). Consequently, in stable socioeconomic conditions it is better to allocate financial resources from operational to capital issues. This contributes to sustainable economic growth. In turn, in pandemic conditions, an excessive expenditure burden on the budget (both current and capital expenditures), on the contrary, creates risks for economic stability. In addition, in the pre-pandemic period, an increase in both public and private expenditures in healthcare is a driver of economic growth. In contrast, in the pandemic period, an increase in out-of-pocket expenditure is a more relevant factor for stimulating sustainable economic growth. Thus, in unstable conditions it is better to reduce capital and current expenditures and increase out-of-pocket expenditure (“out-of-pocket payments are spending on health directly out-of-pocket by households” [41,42]). Consequently, in crisis conditions, healthcare institutions must receive immediate compensation for the cost of the provided medical services in order to provide sufficient medical services.

Both the theoretical and empirical results obtained can be useful to scientists carrying out their research in this direction, to representatives of state authorities of the analysed 15 European countries in the context of policy reform in the field of healthcare and economic development strategy, as well as to the third-sector representatives, analytical companies, and international organisations, when creating explanatory and analytical notes.

Although some interesting findings are obtained in this study, several limitations should be addressed. This study can be deepened by expanding the geographic sample and implementing separate blocks of regression modelling for different clusters of countries characterized by common economic growth rates. Moreover, as in [5,6,7,8,9], it was empirically proven that for some developing countries there are both one-way and bilateral relationships between healthcare expenditure and economic growth; it might become interesting to expand and refresh these research results considering new data. While some research [6,7] argues for the existence of a non-linear relationship between healthcare expenditure and economic growth, it supports the idea about the U-shape causality between healthcare expenditure and economic growth in different periods. Therefore, as a perspective for further empirical research, it might be interesting to clarify the threshold of healthcare expenditure at which economic growth is still boosted and after which it becomes negative. This might be helpful for the development of both economic policy and healthcare policy in certain countries.

## Figures and Tables

**Figure 1 ijerph-20-03049-f001:**
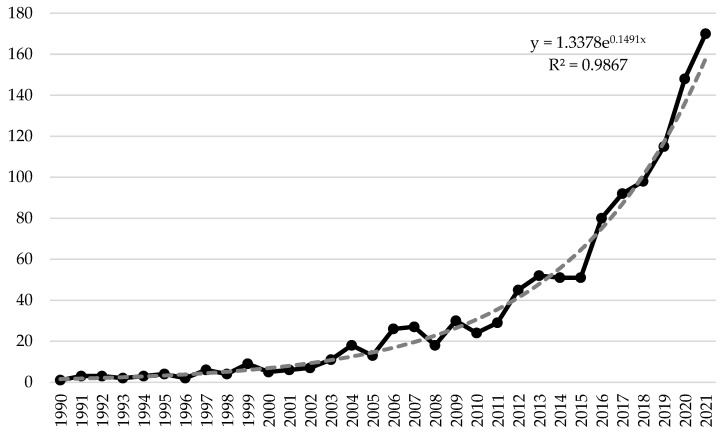
Number of Scopus [4] papers on healthcare expenditure and sustainable growth published during 1990–2021.

**Figure 2 ijerph-20-03049-f002:**
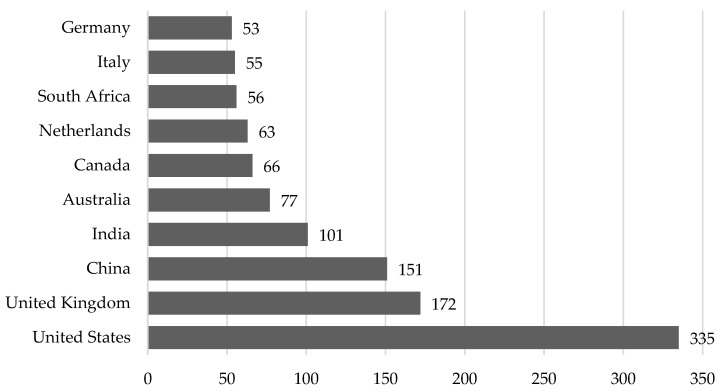
Countries with the most significant amount of Scopus [4] papers on healthcare expenditure and sustainable growth published during 1990–2022 (15 September 2022).

**Figure 3 ijerph-20-03049-f003:**
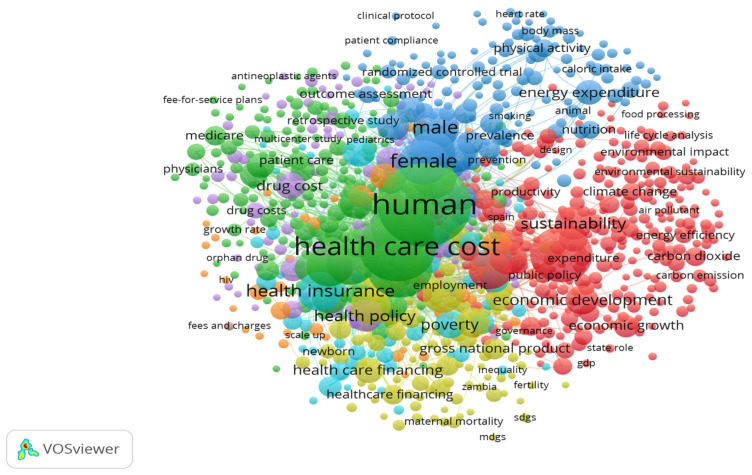
Visualisation map of keywords’ co-occurrence in Scopus [4] publications on healthcare expenditure and sustainable growth realised with VOSviewer_1.6.17 toolkit [3].

**Figure 4 ijerph-20-03049-f004:**
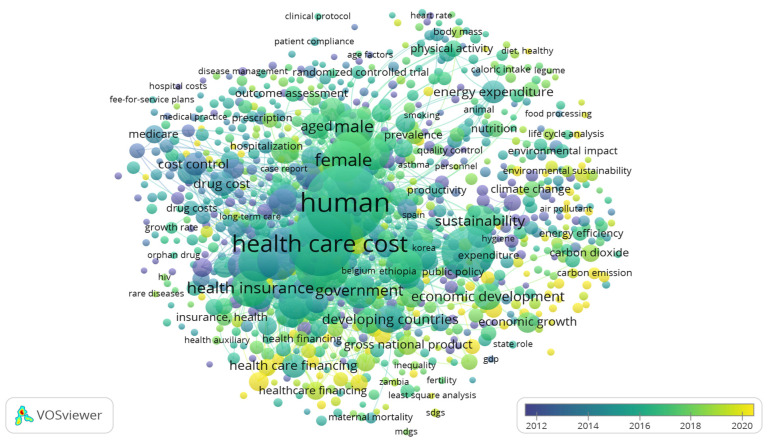
Overlay visualisation map of keyword co-occurrence in Scopus [4] publications on healthcare expenditure and sustainable growth realised with VOSviewer_1.6.17 toolkit [3].

**Table 1 ijerph-20-03049-t001:** Descriptive statistics on sustainable economic growth indicators for 15 European countries in 2000–2021.

Variable	Obs.	Mean	Std. Dev.	Min	Max
SEG_1	330	74.521	3.011	67.01	81.53
SEG_2	330	9.828	4.583	2.01	24.00
SEG_3	306	32.034	5.577	23.20	51.93
SEG_4	330	5.835	2.845	1.03	14.90
SEG_5	330	4128.174	1515.422	1213.12	8085.74
SEG_6	330	3.281	4.229	−15.14	13.94
SEG_7	330	5.560	9.259	−40.08	109.33
SEG_8	330	24.559	5.236	8.93	41.59
SEG_9	217	10.396	5.982	0.01	26.73
SEG_10	330	30.711	13.327	4.47	59.99
SEG_11	330	5.256	8.423	0.03	95.01
SEG_12	230	4.377	4.343	0.47	25.18
SEG_13	330	111.147	32,885	22.49	190.70

Obs.—number of observations; Std. Dev.—standard deviation.

**Table 2 ijerph-20-03049-t002:** Weighting coefficients of sustainable economic growth indicators.

Variable	PCE_1	PCE_2	PCE_3	PCE_4	PCE_5	Average	Rank	WtC
SEG_1	0.273	0.077	0.303	0.057	0.093	0.1606	3	0.0333
SEG_2	0.217	0.363	0.037	0.570	0.041	0.2456	11	0.1222
SEG_3	0.297	0.032	0.231	0.462	0.052	0.2148	6	0.0667
SEG_4	0.329	0.057	0.447	0.292	0.088	0.2426	9	0.1000
SEG_5	0.386	0.089	0.242	0.416	0.080	0.2426	9	0.1000
SEG_6	0.146	0.477	0.169	0.222	0.491	0.3010	13	0.1444
SEG_7	0.016	0.340	0.006	0.284	0.803	0.2898	12	0.1333
SEG_8	0.094	0.631	0.162	0.069	0.127	0.2166	7	0.0778
SEG_9	0.376	0.013	0.357	0.042	0.120	0.1816	5	0.0556
SEG_10	0.385	0.019	0.202	0.122	0.067	0.1590	2	0.0222
SEG_11	0.072	0.216	0.413	0.024	0.178	0.1806	4	0.0444
SEG_12	0.274	0.126	0.446	0.205	0.114	0.2330	8	0.0889
SEG_13	0.365	0.213	0.045	0.075	0.083	0.1562	1	0.0111

PCE—principal component eigenvector; WtC—weighting coefficient.

**Table 3 ijerph-20-03049-t003:** Sustainable Economic Growth Index for 15 European countries in 2000–2021.

	2000	2001	2002	2003	2004	2005	2006	2007	2008	2009	2010	Average
Hungary	0.358	0.358	0.356	0.349	0.358	0.361	0.371	0.449	0.455	0.351	0.365	0.376
Moldova	0.356	0.366	0.372	0.354	0.369	0.377	0.367	0.394	0.432	0.345	0.391	0.375
Estonia	0.331	0.313	0.323	0.331	0.330	0.364	0.419	0.449	0.348	0.268	0.352	0.348
Albania	0.432	0.428	0.428	0.383	0.366	0.430	0.414	0.413	0.457	0.374	0.354	0.407
Czech Republic	0.330	0.329	0.327	0.331	0.329	0.346	0.354	0.403	0.392	0.332	0.370	0.349
Romania	0.330	0.348	0.342	0.331	0.375	0.343	0.388	0.406	0.429	0.341	0.351	0.362
Latvia	0.345	0.348	0.349	0.358	0.365	0.379	0.434	0.445	0.348	0.251	0.306	0.357
Slovak Republic	0.296	0.311	0.326	0.313	0.320	0.331	0.364	0.394	0.367	0.290	0.357	0.333
Slovenia	0.332	0.334	0.338	0.327	0.342	0.337	0.363	0.401	0.393	0.305	0.337	0.346
Lithuania	0.318	0.328	0.339	0.349	0.332	0.347	0.375	0.440	0.370	0.221	0.325	0.340
Poland	0.302	0.273	0.273	0.282	0.292	0.286	0.313	0.339	0.333	0.321	0.325	0.304
Ukraine	0.322	0.344	0.325	0.349	0.360	0.330	0.353	0.371	0.340	0.230	0.330	0.332
Bulgaria	0.255	0.248	0.259	0.265	0.278	0.294	0.358	0.389	0.399	0.318	0.308	0.307
Croatia	0.255	0.254	0.270	0.275	0.263	0.264	0.288	0.310	0.311	0.301	0.314	0.282
Serbia	0.267	0.286	0.284	0.275	0.308	0.287	0.289	0.318	0.318	0.258	0.267	0.287
	**2011**	**2012**	**2013**	**2014**	**2015**	**2016**	**2017**	**2018**	**2019**	**2020**	**2021**	**Average**
Hungary	0.402	0.366	0.368	0.406	0.403	0.457	0.401	0.382	0.509	0.511	0.479	0.426
Moldova	0.397	0.368	0.431	0.444	0.392	0.421	0.431	0.430	0.419	0.383	0.501	0.420
Estonia	0.411	0.402	0.390	0.423	0.403	0.427	0.444	0.447	0.466	0.424	0.483	0.429
Albania	0.337	0.390	0.368	0.359	0.369	0.376	0.366	0.371	0.362	0.327	0.389	0.365
Czech Republic	0.368	0.357	0.356	0.381	0.408	0.407	0.447	0.464	0.479	0.408	0.449	0.411
Romania	0.379	0.372	0.380	0.381	0.382	0.400	0.423	0.415	0.423	0.369	0.418	0.395
Latvia	0.374	0.400	0.425	0.386	0.400	0.378	0.379	0.390	0.384	0.350	0.402	0.388
Slovak Republic	0.348	0.338	0.339	0.360	0.369	0.364	0.379	0.381	0.387	0.332	0.376	0.361
Slovenia	0.333	0.307	0.313	0.343	0.336	0.361	0.356	0.366	0.369	0.345	0.394	0.347
Lithuania	0.361	0.346	0.342	0.353	0.343	0.343	0.357	0.362	0.362	0.320	0.352	0.349
Poland	0.329	0.309	0.306	0.354	0.343	0.345	0.362	0.378	0.384	0.346	0.391	0.350
Ukraine	0.350	0.331	0.329	0.262	0.268	0.342	0.335	0.337	0.330	0.279	0.332	0.318
Bulgaria	0.306	0.301	0.297	0.301	0.330	0.317	0.325	0.337	0.355	0.302	0.340	0.319
Croatia	0.310	0.305	0.324	0.330	0.337	0.353	0.347	0.359	0.374	0.314	0.386	0.340
Serbia	0.279	0.263	0.282	0.263	0.285	0.295	0.297	0.320	0.329	0.303	0.341	0.296

**Table 4 ijerph-20-03049-t004:** Regression results (random-effects GLS regression) on the impact of healthcare expenditure on the Sustainable Economic Growth Index for 15 European countries in the pre-pandemic period (2000–2019).

SEGI	Coef.	St.Err.	t-Value	*p*-Value	95% Confidence Interval		Sig
CapHE	0.05025	0.013	3.85	0.000	0.025	0.076	***
CurHE	−0.01303	0.003	−3.96	0.000	−0.019	−0.007	***
DGGHE	0.01543	0.003	5.96	0.000	0.010	0.021	***
DPHE	0.00509	0.002	3.28	0.001	0.002	0.008	***
OoPE	−0.00258	0.001	−1.72	0.085	−0.006	0.000	*
Constant	0.16426	0.032	5.18	0.000	0.102	0.226	***

*** and * indicate significance at 1% and 10% levels, respectively.

**Table 5 ijerph-20-03049-t005:** Regression results (random-effects GLS regression) on the impact of healthcare expenditure on the Sustainable Economic Growth Index for 15 European countries in the pandemic period (2020–2021).

SEGI	Coef.	St.Err.	t-Value	*p*-Value	95% Confidence Interval	
CapHE	−0.00494	0.084	−0.06	0.953	−0.169	0.160
CurHE	−0.00036	0.024	−0.01	0.988	−0.048	0.047
DGGHE	0.00222	0.019	0.12	0.907	−0.035	0.040
DPHE	−0.00352	0.008	−0.42	0.675	−0.020	0.013
OoPE	0.00172	0.006	0.31	0.759	−0.009	0.013
Constant	0.42500	0.249	1.71	0.087	−0.062	0.912

## Data Availability

Not applicable.

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
