# Peer review of "Impact of Coronavirus Disease COVID-19 on the Relationship between Healthcare Expenditures and Sustainable Economic Growth"

_ijerph, 2023, doi:10.3390/ijerph20043049_

Round 1

Reviewer 1 Report

The article "Impact of coronavirus disease COVID-19 on healthcare expend- 2 itures and sustainable economic growth relationship" (ijerph-2180427) has been an interesting read, although it has to be significantly improved in terms of language. Several sentences have been very difficult to understand and this reduces the easiness to read the article.

Moreover, I think that the Conclusion should better hightlight the relationship between healthcare expenditures and sustainable economic growth, which is not sufficiently clear. In addition, I think that the authors should provide more detailed macroeconomic information on the country sample (15) they took. In fact, they just state that "The country sample was created with consideration of both geographical proximity to Ukraine and Poland, as well as the similarity of economic development trends (in particular, purchasing power standards and income)" while the reader should instead get some data (perhaps, summed up in a specific table).

My suggestion is to revise the paper in terms of language and clarity with a "sentence-by-sentence approach", because currently it is not an easy read which makes its comprehension not particularly simple.

Reviewer 2 Report

Thanks for the opportunity to review this interesting research. I have a few minor comments.

1. Please explain the possible mechanism in which current health expenditure negatively affects SDG index. What do you mean by eating up financial resources? Please elaborate on what kind of items are included in this current expenditure and how it negatively affects SDG index.

2. Table 6 is not promising as all coefficients are statistically insignificant. Compared to Table 5, which used 20 years of data period, Table 6 only covers 2 years. Thus, there is not sufficient year observations for the data analyzed in Table 6.
